# Effect of intraoperative systemic magnesium sulphate on postoperative Richmond Agitation-Sedation Scale score after endovascular repair of aortic aneurysm under general anesthesia: A double-blind, randomized, controlled trial

**Haruna Kanamori, Yoshihito Fujita**[ID]*, **Rina Joko, Ryota Ishihara, Yoshihiro Fujiwara**

Department of Anesthesiology and Intensive Care Medicine, Aichi Medical University School of Medicine, Nagakute, Aichi, Japan

* fujita.yoshihito.823@mail.aichi-med-u.ac.jp, zfjtzz@gmail.com

## Abstract

Intraoperative magnesium has the effect of reducing postoperative opiate requirement, pain, and agitation. However, its effect on postoperative sedation and delirium is unclear. This study investigated the effect of magnesium on the postoperative Richmond Agitation-Sedation Scale (RASS) score and delirium following endovascular repair of aortic aneurysm (EVAR). Sixty-three consecutive patients diagnosed with abdominal (45) and thoracic (18) aortic aneurysm who underwent EVAR under general anesthesia were eligible. Patients were allocated randomly to the magnesium group (infusion of 30 mg•kg$^{-1}$ magnesium in the first hour followed by 10 mg•kg$^{-1}$ h$^{-1}$ until the end of surgical procedure, targeting total 60 mg•kg$^{-1}$) or the control group (0.9% saline at the same volume and rate). The primary outcome was whether magnesium had an effect on RASS score of patients at postoperative ICU admission. Secondary outcomes were effects on RASS score, numerical rating scale (NRS) score, Confusion Assessment Method for the Intensive Care Unit (CAM-ICU) until 24 h after postoperative ICU transfer, and length of ICU stay. At postoperative ICU admission, magnesium had no significant effect on the RASS score (0[−0.5 to 0] vs 0[0 to 0]; P = 0.114), but at 1 h the NRS score was statistically different, 2[0 to 4] vs 4[0 to 5] (P = 0.0406). However, other data (RASS score, NRS score, CAM-ICU and length of ICU stay) did not show a significant difference. Our results did not show that intraoperative magnesium of target total 60 mg•kg$^{-1}$ affected postoperative RASS score for undergoing EVAR.

**Trial registration:** The current study was registered according to WHO and ICMJE standards on 4 July 2018, under registration number the Japan Registry of Clinical Trials, iRCTs041190013.

**Data Availability Statement:** All relevant data are within the paper and its Supporting information files.

**Funding:** The authors received no specific funding for this work.

**Competing interests:** The authors have declared that no competing interests exist.

## Introduction

Delirium is common in vascular surgery settings [1, 2], and associated anesthesia management and intra-operative drugs have attracted substantial interest [3, 4]. Intraoperative magnesium has the effect of reducing intra-operative and postoperative opiate requirement and pain [5–8]. Recently it was reported to decrease postoperative agitation [9], which might suggest that magnesium could have an effect on not only pain relief but also sedative level. To our knowledge, for the first time, that paper showed possibility of magnesium which has showed pain relief also has an effect on delirium and sedative level. However, its effect on postoperative sedation and delirium is unclear [8]. We investigated its effect on the postoperative Richmond Agitation-Sedation Scale (RASS) score, pain score (numerical rating scale: NRS), delirium (Confusion Assessment Method for the Intensive Care Unit: CAM-ICU), and length of ICU stay. Therefore, we hypothesized that magnesium infusion would reduce postoperative Richmond Agitation-Sedation Scale (RASS) score after surgery for endovascular aortic aneurysm repair (EVAR) and the primary outcome was an effect on the RASS scores and the second outcomes were effects on the incidence of delirium (Confusion Assessment Method for the Intensive Care Unit: CAM-ICU), pain score (numerical rating scale: NRS), and length of ICU stay.

## Materials and methods

Ethical approval for this study (Ethical Committee CRB4180011) was provided by the Ethical Committee of Aichi Medical University Hospitals, Nagakute, Japan (Chairperson Prof. Masayuki Hanyuda) on 18 October 2018. The current study was registered according to WHO and ICMJE standards under the registration number Japan controlled jRCTs041190013. In addition, written informed consent was obtained from all participants. With our misunderstanding of the procedure of jRCT registration, difference of the registration between our IRB and jRCT on the WEB (in detail in S1 File) happened. However, the authors confirmed that all ongoing and related trials for this drug and intervention were registered. And we started our recruiting the first participant in our study after getting registration for our study.

Patients of either sex aged 60 years or older who were scheduled for endovascular repair of abdominal or thoracic aortic aneurysm under general anesthesia at our hospital were eligible. The reason why we chose aged 60 years or older was not to decrease the incidence of delirium, Exclusion criteria were: (1) patients who required emergency surgery; (2) patients with severe complications, including cardiac, renal (including dialysis), blood, lung, liver, or life-threatening disease; (3) patients with serious complications arising from psychological illness (including bipolar disorder, suicide intent). Severe renal dysfunction was defined as estimated glomerular filtration rate (eGFR) less than 30 ml•min$^{-1}$•1.7 m$^{-2}$ or utilization of hemodialysis. The first patient entered the study from 26 October 2018.

Patients were assigned randomly to one of two groups to receive intra-operative magnesium (magnesium sulphate; Otsuka Pharmaceutical, Japan) (magnesium group) or 0.9% saline solution (control group). A computer-generated randomization program "Research randomizer" from on the Web was used and randomization was 1:1 ratio. The independent anesthesiologist who was not involved in anesthesia management and outcome evaluation provided a pharmacist with sealed envelopes which included patient identification, group allocation and body weight, and the attending pharmacist prepared the colorless coded solutions in transparent syringes. The codes were kept confidential by the independent anesthesiologist until completion of the study. The pharmacist prepared the magnesium-solution syringes by mixing magnesium sulphate with 0.9% saline according to the actual body weight or ideal body weight. To avoid overdose of magnesium administration to obese patients, we applied ideal body weight (body mass index: 22) when the actual body weight was greater than the ideal

one. We used magnesium sulphate solution containing 123 mg of magnesium per 1 ml. For each patient, a total dose of magnesium sulphate of 60 mg•kg$^{-1}$ was diluted to 60 ml. Patients in the magnesium group received an initial intravenous loading dose of 30 mg•kg$^{-1}$ over 1 h, followed by a continuous infusion of 10 mg•kg$^{-1}$ h$^{-1}$ for the duration of surgery. The infusion was terminated when the duration exceeded 4 h in total, and was delivered using a syringe pump. The control group received an equivalent volume of 0.9% saline.

No patient received premedication. On arrival at the operating room, standard monitoring of pulse oximetry (SpO$_2$), noninvasive blood pressure (NIBP), ECG, and heart rate (HR) was implemented. An acceleromyograph (TOF-Watch SX; Organon, Ireland) was attached to stimulate either ulnar or facial nerve to measure the response of adductor pollicis or corrugator supercilii, respectively, to monitor neuromuscular block, because the administration of magnesium can extend the effect of non-depolarizing neuromuscular relaxants. The Bispectral Index (BIS) was also monitored by an Aspect XP A2000 device (Aspect Medical Systems, Minneapolis, MN, USA) to prevent awareness, a possible contributing factor to postoperative agitation.

Anesthesia was induced with propofol 0.5 to 2 mg•kg$^{-1}$, remifentanil 0 to 0.3 μg•kg$^{-1}$•min$^{-1}$, and fentanyl 0 to 200 mg, followed by rocuronium 0.6 to 0.9 mg•kg$^{-1}$ to facilitate tracheal intubation. Anesthesia was maintained with an inspired desflurane in air/O$_2$ mixture. During the operation, rocuronium 10 mg was administrated when the patient showed two twitch responses of train-of-four (TOF) stimuli. The anesthesiologist adjusted the concentration of desflurane to maintain the BIS value between 40 and 60. Mean arterial blood pressure was monitored continuously and maintained within 70% and 130% of the value before induction of anesthesia. Bolus injection of ephedrine or phenylephrine was used to treat hypotension, which was less than 90 mmHg of systolic blood pressure. Bolus injection of nicardipine hydrochloride was used to treat hypertension, which was more than 140 mmHg of systolic blood pressure. In this study we did not provide a strategy of permissive hypotension. If the attending anesthesiologist needed drugs for postoperative analgesia, acetaminophen 1,000 mg and/or flurbiprofen axetil 50 mg was given intravenously during the operation.

All the EVAR surgeries were performed with radiological imaging in a hybrid operating room attended by not only vascular surgeons but also radiologists who were specialists in this field, thus ensuring that all patients obtained sufficient cerebral blood perfusion during surgery. For anticoagulation, routinely 3,000 units of intravenous unfractionated heparin were administered during surgery, and the administered heparin was titrated to maintain 150 to 200 seconds of activated coagulation time. In our EVAR surgeries there was no use of additional oral anticoagulation drugs before and after surgery. However, if the patients used oral anticoagulation drugs before surgery for any reason, the drugs were stopped before surgery and restarted 24 h after surgery. As for a radiological contrast, the minimum volume of a radiological contrast was administered to perform surgical procedure by vascular surgeons and radiologists.

The infusion of magnesium or saline was terminated when the operator started to close the wound. We set the upper limit of the total infusion period to 4 h for 60 mg•kg$^{-1}$. At the end of surgery, sugammadex was given intravenously to reverse residual neuromuscular block. When the attending anesthesiologist assessed that the patient had recovered sufficiently from anesthesia, the tracheal tube was removed. Patients were then transferred to the postoperative ICU.

We recorded total doses of fentanyl, remifentanil and magnesium sulphate, duration of anesthesia, duration of operation, amount of bleeding, amount of crystalloid fluid infusion (contains Mg 1 mmol•l$^{-1}$), total plasma magnesium value, and ionized plasma magnesium value, using an ion-selective analyzer (Stat Profile Prime ES Comp Analyzer; NOVA

Biomedical, Waltham, MA, USA) before induction of anesthesia, at the ICU admission, and on the day after surgery at 6 a.m.

All patients were extubated at the operating room and were moved to the ICU. And, as soon as the patients arrived at the ICU, the ICU nurses examined vital signs and evaluated RASS, NRS and CAM-ICU. Discharge criteria from the ICU were stable vital signs, free of major complications that require intensive care. Delirium was defined as CAM-ICU positive. RASS score (0, alert and calm; +1, restless; +2, agitated; +3, very agitated; +4, combative; −1, drowsy; −2, light sedation; −3, moderate sedation; −4, deep sedation; −5, unrousable1) was assessed in the ICU by a blinded observer, an attending ICU nurse. Pain scores were assessed using an NRS pain score as rated by the patient, where 0 represents no pain and 10 is the worst imaginable pain. Blinded nurses in the ICU assessed and recorded CAM-ICU, RASS and NRS at the postoperative ICU admission (time 0 h) and 1 h, 6 h and 24 h after the admission of ICU. If the patient was discharged from the ICU within 24 h, the nurse recorded those values of sedative level at the time of ICU discharge. Usage of acetaminophen and dexmedetomidine, and the length of stay in ICU, were also recorded.

Primary outcome of this study was postoperative RASS score at the postoperative ICU admission. Secondary outcomes were incidence of delirium defined using CAM-ICU, pain according to NRS, use of analgesic drugs, which were during ICU stay and length of ICU stay. Therefore, attending surgeons, attending anesthesiologists, attending operative staff, attending ICU staff, and outcome evaluators could never know which group to which the patient belonged.

## Statistical analysis

We calculated the sample size for the primary analysis on the basis of differences seen in the previous study. In the study about reducing postoperative agitation in patients undergoing functional endoscopic sinus surgery (FESS), the difference in RASS scores between two groups was 1 to 2 during the observation period.[9] In this previous study, the SD was not presented. In another retrospective observational study (not published), we investigated the rate of hyperactive delirium among patients who underwent EVAR surgery under general anesthesia in our hospital in the same setting. We reported RASS scores of 0.25±0.5 and 58 cases of hyperactive delirium among 237 patients. Therefore, we estimated that score levels differed by 1 to 2 with SD of 0.5 to 1. Assuming an α level of 0.05 and 90% power, the required number of patients for each group to observe a RASS difference is 22 at most. We considered that a total workable sample size would be 30 patients to detect a difference of 1 to 2, assuming a two-tailed type I error of 5% and type II error of 10%. To allow for a 5% dropout rate, we randomly assigned 32 patients.

Data are expressed as the mean±SD or median [IQR] for non-normally distributed variables (Kolmogorov–Smirnov test), or number and percentage as appropriate. All statistical test were performed two sided. *P* values of less than 0.05 were considered significant. Quantitative variables were compared using Student's *t* test or the Mann–Whitney *U* test where appropriate. Categorical variables are described using number (%) and were compared using χ2 test or Fisher's exact test. Interim analysis was scheduled to take place after recording results from half of the sample size, at which point the decision to continue or stop the study would be made on the basis of valid reasons. For the calculated sample size to be deemed insufficient such reasons would include the potential magnitude of bias or the potential impact on interpretation of the results.

All analyses were performed by EZR version 1.52 (Saitama Medical Centre, Jichi Medical University, Saitama, Japan), which is a graphical interface for R version 4.02 (The R

Foundation for Statistical Computing, Vienna, Austria). More precisely, it is a modified version of R commander designed to add statistical functions frequently used in biostatistics [10].

## Results

One hundred and twenty-four patients were approached to take part in this study, of whom 64 were successfully recruited and 63 included for data analysis. There were 31 patients in the magnesium group and 32 patients in the control group. Fig 1 shows the flow diagram for the study protocol. Table 1 shows the baseline demographic characteristics and patient data.

There was no significant effect on RASS score (0 [0 to 0] vs 0 [0 to 0]; P = 0.114) at postoperative ICU admission, time 0 hour (Table 2). In addition, at other times during the study period (at 1, 6, 12, and 24 h), magnesium had no significant effect on the RASS score. The NRS score was statistically different, 0 [0 to 3] vs 3 [0.5 to 6.5] (P = 0.048) at 1 h. However, during the study period, the data at other times (at 0, 6, 12 and 24 h) did not show significant differences. In addition, delirium estimated by CAM-ICU showed no change of incidence with magnesium administration during the study period, and there was no difference in the length of ICU stay.

We measured total magnesium and ionized magnesium pre-operatively, at postoperative ICU admission, and on the next morning at 6 a.m. (Table 3). Measured ionized magnesium was significantly different (0.66±0.10 vs 0.50±0.13 mmol•l$^{-1}$, $P$<0.0001) at postoperative ICU admission. In addition, on the next morning at 6 a.m., the significant difference continued

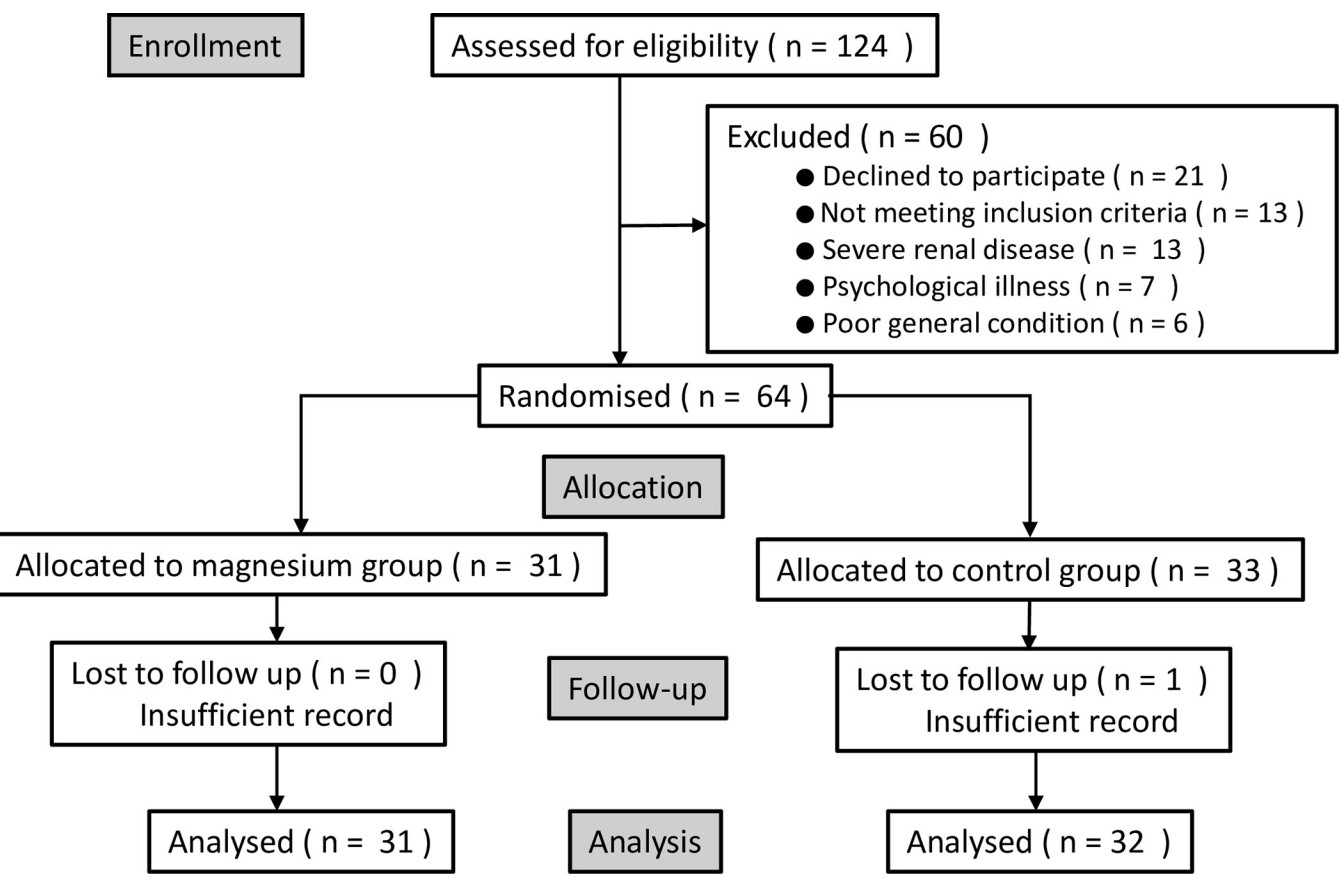

**Fig 1. CONSORT study flow diagram.**

**Table 1. Patients' characteristics.**

| | Magnesium (*n* = 31) | Control (*n* = 32) |
|---|---|---|
| Sex: M: F | 31:1 (96.8:3.2) | 26:6 (81.3:18.8) |
| Age (years) | 76 [71 to 78] | 77 [73 to 82.3] |
| Height (cm) | 166 [160 to 168] | 161 [158 to 165] |
| Weight (kg) | 61 [58 to 63.4] | 61 [54.5 to 66.3] |
| eGFR (ml•min$^{-1}$•1.7 m$^{-2}$) | 58 [50 to 71] | 65 [46.3 to 74.3] |
| Clinical characteristics | | |
| Hypertension | 19 (58.1) | 27 (84.4) |
| Diabetes mellitus | 5 (16.13) | 5 (15.62) |
| Asthma | 1 (3.23) | 2 (6.25) |
| Allergy | 5 (16.1) | 6 (18.8) |
| History of stroke | 0 (0) | 4 (12.5) |
| Smoking | 18 (56.1) | 16 (50) |
| Surgery | | |
| Thoracic aortic aneurysm | 9 (29.0) | 9 (38.1) |
| Abdominal aortic aneurysm | 22 (71.0) | 23 (71.9) |
| Duration of anaesthesia (min) | 204 [173 to 231] | 200 [172.5 to 228.5] |
| Duration of surgery (min) | 148 [125 to 176] | 151.5 [119 to 174.8] |
| Fentanyl (µg) | 200 [200 to 250] | 200 [200 to 250] |
| Remifentanil (mg) | 0.8 [0.6 to 1.05] | 0.75 [0.6 to 1] |
| Acetaminophen (mg) | 1000 [1000 to 1000] | 1000 [1000 to 1000] |
| Flurbiprofen (mg) | 50 [0 to 50] | 50 [0 to 50] |
| Crystalloids (ml) | 1200 [900 to 1500] | 1025 [850 to 1312.5] |
| Blood loss (g) | 57 [38 to 107] | 65 [23.5 to 144.8] |
| Length of ICU stay (days) | 1 [1 to 1] | 1 [1 to 1] |

Data are shown as median [IQR] or number (%).

BMI, body mass index; eGFR, estimated glomerular filtration rate; ICU, Intensive Care Unit.

(0.58±0.12 vs 0.49±0.04 mmol•l$^{-1}$, *P*<0.0001). We observed the same tendency in the total magnesium (2.86±0.28 vs 1.93±0.16 mg•ml$^{-1}$ at postoperative ICU admission, P<0.0001; and 2.2±0.15 vs 2.0±0.16 mg•ml$^{-1}$ next morning, *P*<0.0001).

During the observational period, no adverse events caused by magnesium infusion were recorded, which were existence or non-existence of arrythmia, bradycardia, hypotension, effects of muscle paralysis, and convulsion.

## Discussion

Intraoperative systemic magnesium of target total 60 mg•kg$^{-1}$ did not show an effect on post-operative RASS score and delirium after EVAR surgery in this randomized controlled study. We expected magnesium to have a beneficial neurological effect on delirium as well as on pain relief, because magnesium is purported to have various effects [11], and its effect on agitation has been shown recently [9]. However, the effect of magnesium on delirium has remained unclear to date and there are few studies on this topic, which was sufficient reason for Ng et al. to abandon their systematic review of the effect of magnesium on morphine requirement and investigate the potential neurological effect of magnesium on the reduction of delirium or agitation in surgical patients [8, 12]. In our setting, intraoperative systemic magnesium did not

**Table 2. Postoperative outcome.**

| | Postoperative ICU admission (0h) | 1 h | 6 h | 12 h | 24 h |
|---|---|---|---|---|---|
| RASS | | | | | |
| Magnesium group | 0 [−0.5 to 0] | 0 [0 to 0] | 0 [0 to 0] | 0 [0 to 0] | 0 [0 to 0] |
| Control group | 0 [0 to 0] | 0 [0 to 0] | 0 [0 to 0] | 0 [0 to 0] | 0 [0 to 0] |
| P | 0.114 | 0.238 | 0.347 | 0.621 | 0.518 |
| NRS | | | | | |
| Magnesium group | 0 [0 to 3] | 2 [0 to 4] | 1 [0 to 2] | 0 [0 to 1] | 0 [0 to 1] |
| Control group | 3 [0 to 6] | 4 [0 to 5] | 2 [0 to 3] | 1 [0 to 2] | 1 [0 to 1.5] |
| P | 0.0612 | 0.0406* | 0.14 | 0.498 | 0.226 |
| CAM-ICU positive | | | | | |
| Magnesium group (n = 31) | 2 | 0 | 1 | 1 | 0 |
| Control group (n = 32) | 2 | 1 | 0 | 0 | 0 |
| P | 1 | 1 | 1 | 1 | 1 |
| Postoperative use of analgesics | | | | | |
| Acetaminophen | | | | | |
| Magnesium group | 8/31 | | | | |
| Control group | 11 / 32 | | | | |
| P | 0.582 | | | | |
| Dexmedetomidine | | | | | |
| Magnesium group | 6 / 31 | | | | |
| Control group | 7 / 32 | | | | |
| P | 1 | | | | |

RASS, Richmond Agitation-Sedation Scale; NRS, Numerical Rating Scale pain.

CAM-ICU, Confusion Assessment Method for the Intensive Care Unit.

Data are shown as median [IQR] or number.

*Statistically significant.

demonstrate a significant effect on postoperative RASS score and delirium. We considered two main reasons for these results.

First, an administered magnesium target total of 60 mg•kg$^{-1}$, actually 48±8 mg•kg$^{-1}$, might not be enough to obtain good outcomes. We used the previous protocol which demonstrated in the FESS study [9]. Many protocols for magnesium administration have tried to reduce analgesics consumption and pain relief [13]. Even a low dosage magnesium was effective for

**Table 3. Ionized magnesium concentrations.**

| | Mg administration (mg•kg$^{-1}$) | Mg concentration (mmol•l$^{-1}$) | | |
|---|---|---|---|---|
| | | Before infusion | At postoperative ICU admission | Next morning |
| Total magnesium (mg ml$^{-1}$) | | | | |
| Magnesium group | 48.32±8.1 | 2.0±0.23 | 2.86±0.28 | 2.2±0.15 |
| Control group | 0 | 1.9±0.19 | 1.93±0.16 | 2.0±0.16 |
| P | | 0.068 | <0.0001 | <0.0001 |
| Ionised magnesium (mmol l$^{-1}$) | | | | |
| Magnesium group | 48.32±8.1 | 0.53±0.091 | 0.66±0.095 | 0.58±0.12 |
| Control group | 0 | 0.48±0.051 | 0.50±0.13 | 0.49±0.043 |
| P | | 0.012 | <0.0001 | <0.0001 |

Data are shown as mean ± SD.

pain relief in laparoscopic cholecystectomy [14]. However, a high-dose magnesium was administered for the duration of mastectomy, laparoscopic gastrectomy, and video-assisted thoracoscopic surgery [15–17]. Our protocol of magnesium administration may be comparatively low for obtaining good results. To obtain a neurological effect, we might have to follow a higher-dose protocol than that employed in the present study.

The optimal target level of ionized magnesium is unclear. In the field of obstetric intensive care, magnesium represents the first-choice medication in treatment and prevention of eclamptic seizures at a therapeutic level of 2.0 to 3.5 mmol•l$^{-1}$ [18]. In addition, in one study with a higher-dose setting, the magnesium concentrations were 1.25±0.28 mmol•l$^{-1}$ after magnesium infusion during mastectomy [15]. Moreover, this study demonstrated the positive correlation between magnesium concentration and postoperative quality-of-recovery scores. We might have to target a higher level of ionized magnesium to demonstrate a favorable neurological effect.

Second, we suspect that one of the reasons for our negative outcome is the rare incidence of RASS score >0 and delirium. Before the study we calculated the sample size from our previous data, which showed that we had RASS scores of 0.25±0.5 and 58 cases (24.5%) of hyperactive delirium in 237 patients (not published). However, in the current study we had only six occurrences of RASS score >0 out of 315 observations, six cases of positive CAM-ICU out of 315 observations, and a total of only four (6.3%) patients with positive CAM-ICU. We chose EVAR surgeries for this study because of the high incidence (24.5%) of delirium in our previous study. As a consequence, we encountered very low incidence. Therefore, we might not be able to detect positive neurological effects in this study and the sample size might be small to detect statistical significance. We were unable to clearly explain this change of incidence. In any event, in our setting we did not record sufficient incidence to detect a significant effect of magnesium on RASS scores and delirium.

We chose RASS scores as primary outcomes. RASS score was a surrogate for reducing delirium incidence, not a clinical outcome goal in itself. The modified RASS has good sensitivity and specificity for incident delirium [19]. In addition, scores might be likely to yield a significant difference compared with an all-or-none outcome, so we chose RASS score difference for this study. If there is a sufficient number of cases from which to obtain a statistical difference, delirium incidence should be the primary outcome. Therefore, in the future we should plan joint research with other facilities to expand the cohorts.

Our study was performed in a strict, randomized, double-blind controlled fashion, which was one of the strengths of the study, although magnesium infusion did not show a neurological effect in this setting. However, this does not mean that there was no neurological effect on sedated condition and delirium, because these two factors should affect the outcome. We should gather data in a setting where substantial incidence of delirium can be obtained. In addition, a sufficiently high level of ionized magnesium should be allocated to the magnesium group. One study showed the dose–response effect of magnesium on postoperative pain relief [15], and nowadays ionized magnesium can easily be measured at the bedside. Therefore, we also believe that future study investigating the dose–response effect of magnesium could provide valuable information about the optimal or sufficient dosage of magnesium to achieve a steady-state outcome [8].

There are some limitations to the current study. First, the limited sample size from a single center may limit the generalizability of our findings. In addition, our small simple size was calculated with our previous data that had showed many cases of delirium and in the interim analysis our study could have shown the statistical significance, however, finally, we could not show the statistical significance. We should have more careful calculation in sample size and should have taken more numbers in the sample size. Second, the incidence of delirium

decreased dramatically in this study compared with our previous observational data. We could not find precise reasons for this, and a reasonable explanation remains lacking. In any event, in this study the incidence was too low for a neurological effect of magnesium to be detected. Third, we did not analyze the dose–response relationship between amount of magnesium infused and the concentration. In fact, the ionized magnesium of administered magnesium did vary; however, we did not have sufficient samples in this setting from which to estimate the dose–response relationship. Future study should investigate further the dose–response issue.

In conclusion, our results did not show that intra-operative magnesium to a target total of 60 mg•kg$^{-1}$ had an effect on postoperative RASS score and delirium. For magnesium to obtain a neurological effect, a sufficiently high level of ionized magnesium might be required.

## Supporting information

**S1 File.**
(DOCX)

**S2 File.**
(DOCX)

**S3 File.**
(DOCX)

**S4 File.**
(DOCX)

**S1 Data.**
(XLSX)

**S1 Checklist. CONSORT 2010 checklist of information to include when reporting a randomised trial**[*].
(DOC)

## Acknowledgments

Assistance with the study: the authors thank our anesthesia department staff for anesthesia management, pharmacologists in our hospital for making blinded drugs, and general ICU nurses for postoperative management and evaluation of postoperative conditions. We thank Hugh McGonigle, from Edanz Group (https://en-author-services.edanz.com/ac), for editing a draft of the manuscript.

**Presentation:** We presented a part of this article at Euro Anaesthesia 2020.

## Author Contributions

**Conceptualization:** Haruna Kanamori, Yoshihito Fujita, Yoshihiro Fujiwara.

**Data curation:** Haruna Kanamori, Yoshihito Fujita, Rina Joko, Ryota Ishihara.

**Formal analysis:** Haruna Kanamori, Yoshihito Fujita, Ryota Ishihara.

**Investigation:** Haruna Kanamori, Yoshihito Fujita, Rina Joko, Ryota Ishihara.

**Methodology:** Haruna Kanamori, Yoshihito Fujita, Rina Joko, Ryota Ishihara.

**Project administration:** Yoshihito Fujita, Rina Joko, Ryota Ishihara.

**Resources:** Yoshihito Fujita.

**Software:** Yoshihito Fujita.

**Supervision:** Yoshihiro Fujiwara.

**Validation:** Haruna Kanamori, Yoshihito Fujita, Rina Joko, Ryota Ishihara, Yoshihiro Fujiwara.

**Visualization:** Yoshihito Fujita, Yoshihiro Fujiwara.

**Writing – original draft:** Haruna Kanamori, Yoshihito Fujita.

**Writing – review & editing:** Yoshihito Fujita, Yoshihiro Fujiwara.

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
