## [Decision Letter · Decision Letter 0]

12 May 2022

PONE-D-22-03920Effect of intraoperative systemic magnesium sulphate on postoperative Richmond Agitation-Sedation Scale score after endovascular repair of aortic aneurysm: a double-blind, randomized, controlled trialPLOS ONE

Dear Dr. Fujita,

Thank you for submitting your manuscript to PLOS ONE. After careful consideration, we feel that it has merit but does not fully meet PLOS ONE’s publication criteria as it currently stands. Therefore, we invite you to submit a revised version of the manuscript that addresses the points raised during the review process. All reviewers expressed concerns with respect to the sample size. Given the fact that the actual power of your study is dictated by its confidence intervals I would suggest that you pay particular attention the implications of your trial. If the conclusion is that the power of the trial is too low for any conclusions with respect to outcomes, I would suggest that you pay particular attention to the lessons can be learned for future studies/study designs.

We look forward to receiving your revised manuscript.

Kind regards,

Jan H.N. Lindeman

Academic Editor

PLOS ONE

Journal Requirements:

2. Registration done retrospectively (after enrollment of participants) (TC2/PRTC Note)

Thank you for submitting your clinical trial to PLOS ONE and for providing the name of the registry and the registration number. The information in the registry entry suggests that your trial was registered after patient recruitment began. PLOS ONE strongly encourages authors to register all trials before recruiting the first participant in a study.

1) your reasons for your delay in registering this study (after enrolment of participants started);

2) confirmation that all related trials are registered by stating: “The authors confirm that all ongoing and related trials for this drug/intervention are registered”

Reviewers' comments:

Reviewer's Responses to Questions

**Comments to the Author**

1. Is the manuscript technically sound, and do the data support the conclusions?

Reviewer #1: Yes

Reviewer #2: Partly

Reviewer #3: Yes

2. Has the statistical analysis been performed appropriately and rigorously? 

Reviewer #1: Yes

Reviewer #2: No

Reviewer #3: Yes

3. Have the authors made all data underlying the findings in their manuscript fully available?

Reviewer #1: Yes

Reviewer #2: Yes

Reviewer #3: Yes

4. Is the manuscript presented in an intelligible fashion and written in standard English?

Reviewer #1: No

Reviewer #2: Yes

Reviewer #3: Yes

5. Review Comments to the Author

Reviewer #1: I would like to congratulates the authors for having successfully conducted a well-prepared RCT. This is a cross topic involving anesthesia and vascular surgery, and it seems that few studies to date have investigated this topic. Though negative results yielded, the study is still considered of good value for understanding the impact of magnesium on vascular surgery.

I have the following questions and suggestions:

1. The identifier of trial registration is different in title page(jRCTs041190013) and Methods and Materials (iRCTs041190013), please check.

2. At page 5, “The first patient entered the study from 26 October 2018 to 15 July 2020. ” caused confusion, please check and revise.

3. At page 5, “A computer-generated randomization programme was used. ”. This should be described in detail, what software did you use? Or what is the block size of randomization?

4. From page 6 to 9, you described of performing anesthesia, performing surgery, and transferring patients to ICU. This part is too long, please make it concise but understandable.

5. At page 11, the determination of sample size is questionable. The data of sinus surgery and your unpublished data was combined to calculate sample size, which undoubtably caused imprecision. If this is not amendable, please explain in the limitation section.

6. In the discussion section, you mentioned that the “magnesium target total of 60 mg•kg−1, actually 48±8 mg•kg−1, was not enough to obtain good outcomes ” and “Our protocol of magnesium administration was comparatively low. ”.

This could really be misleading, because your dosage is not low! Your dosage of “ 48±8 mg•kg−1” is actually higher that most magnesium trials [as in PMID: 17513654]. This rationale should be reconsidered.

7. In the discussion section, you mentioned that the the incidence of agitation and delirium were low. This contradicts to your sample size calculation. If you have known the agitation and delirium is rare, would you used your current sample size calculating method?

I understand that what’s done is hard to change, please think of a way to make this more logical.

8. The language is poor and not easy to understand, please seek professional assistance.

Reviewer #2: This is an RCT looking at the effect of magnesium infusion on post-operative pain relief and sedative level after surgery for endovascular aortic aneurysm repair (EVAR).

1) It would be beneficial for the manuscript if the authors can give more details in the introduction. For example just mentioning that “effect on postoperative sedation and delirium is unclear” might not be sufficient. Consider what study designs have been used previously to justify using an RCT for the current study, this would be good to know. For example, there is mention of a retrospective study in the sample size calculation (although this is unpublished).

2) Additionally considering the target patient population is aged 60 and above, a reason/rationale in the introduction relating to the commonality of pain and delirium in this target group would add more context to the study.

3) The introduction should explicitly state the primary and secondary objectives.

4) Where safety outcomes considered?

5) Can authors also mention how missing data will be handled.

6) It would be beneficial for the manuscript if the authors can define population of analysis (e.g ITT, per-protocol, safety)

7) Did that interim that was scheduled to take place happen and if it did, was this accounted for in the power calculation to account for the type I error, if the interim did/didnt happen can this be stated in the manuscript.

8) For randomisation, explicitly state that it was 1:1 ratio.

9) Although secondary outcomes have stated, adding information relating to repeated measures would add more information.

10) The choice of words may need to be considered, i.e secondary outcomes as incidence, implies you are looking at incidence in the context of epidemiology (defined as number of cased in a defined period). This would mean you would need to report incidence per person time follow-up. Which this is not the case. Perhaps consider revising this.

11) Can the final couple of sentences for sample size explicitly state which SD was considered.

12) The analysis is based on multiple time-points how was this accounted for in the analysis, multiple testing.

Reviewer #3: Well conducted randomized study looking at the effects of magnesium on RASS, NPS, and CAM-ICU.

Although negative studies are valuable to the literature, this negative study is difficult to really incorporate into our understanding of the effects of magnesium. The study size was very small and as the authors point out, the power of this study was just 0.416. This also is only a single center study looking at a single surgical case with relatively low pain issues. It doesn't seem that the magnesium contributed to any changes in intra-operative or post-operative narcotic use.

6. PLOS authors have the option to publish the peer review history of their article (what does this mean?). If published, this will include your full peer review and any attached files.

Reviewer #1: No

Reviewer #2: No

Reviewer #3: No

---

## [Author Response · Author response to Decision Letter 0]

29 Jun 2022

Professor Jan H.N. Lindeman

Academic Editor

PLOS ONE

June 19, 2022

Dear Professor Younsuk Lee,

Re: Manuscript reference No. PONE-D-22-03920

Please find attached a revised version of our manuscript “Effect of intraoperative systemic magnesium sulphate on postoperative Richmond Agitation-Sedation Scale score after endovascular repair of aortic aneurysm: a double-blind, randomized, controlled trial”, which we would like to resubmit for publication as a clinical trial to the Academic Editor in PLOS ONE. 

The comments of the reviewers were highly insightful. In the following pages are our point-by-point responses to each of the comments of the reviewers.

Revisions in the text are shown redlined. We hope that the revisions in the manuscript and our accompanying responses will be sufficient to make our manuscript suitable for publication in PLOS ONE.

We shall look forward to hearing from you at your earliest convenience.

Yours sincerely,

Yoshihito Fujita, MD, PhD

Address: 1-1 Karimata Yazako Nagakute, Aichi, Japan

Ph: +81-561-62-3311

Fax: +81-561-62-4866

E-mail: fujita.yoshihito.823@mail.aichi-med-u.ac.jp

Response to comments of the Editor:

Response: Thank you for the important point for our manuscript. We tried to rewrite the content according to your suggestions. We checked our manuscript according to the PLOS ONE’s style. 

2. Registration done retrospectively (after enrollment of participants) (TC2/PRTC Note)

Thank you for submitting your clinical trial to PLOS ONE and for providing the name of the registry and the registration number. The information in the registry entry suggests that your trial was registered after patient recruitment began. PLOS ONE strongly encourages authors to register all trials before recruiting the first participant in a study.

1) your reasons for your delay in registering this study (after enrolment of participants started);

2) confirmation that all related trials are registered by stating: “The authors confirm that all ongoing and related trials for this drug/intervention are registered”

Response: Thank you for the important point for our manuscript. We are really sorry for this failure of the registration procedure. As the letter I sent to PLOS ONE, we misunderstood that we had finished all registration procedures with the meeting with jRCT staffs and we got the approved paper on 18 October 2018 from our university IRB. However, actually, we should have sent the documents that we should have printed out from WEB site, to the jRCT and after getting the letter from us, the procedure would have done and jRCT would have registered our study on the WEB. We thought that our protocol had already been approved, therefore, we started patient recruitment. We did not notice that on jRCT WEB our protocol had not approved yet. As soon as possible after noticing it, we tried to confirm the popper procedure and send the needed documents to jRCT. As the results, the difference between the text and URL on the registration dates happened. 

We added the lower sentences to the text (page 5, line 2)and add the supplement 1 that included of the reasons mentioned in detail. 

With our misunderstanding of the procedure of jRCT registration, difference of the registration between our IRB and jRCT on the WEB (in detail in supplement 1) happened. However, the authors confirmed that all ongoing and related trials for this drug and intervention were registered. And we started our recruiting the first participant in our study after getting registration for our study.

Reviewers' comments:

Reviewer's Responses to Questions

Comments to the Author

1. Is the manuscript technically sound, and do the data support the conclusions?

Reviewer #1: Yes

Reviewer #2: Partly

Reviewer #3: Yes

2. Has the statistical analysis been performed appropriately and rigorously?

Reviewer #1: Yes

Reviewer #2: No

Reviewer #3: Yes

3. Have the authors made all data underlying the findings in their manuscript fully available?

Reviewer #1: Yes

Reviewer #2: Yes

Reviewer #3: Yes

4. Is the manuscript presented in an intelligible fashion and written in standard English?

Reviewer #1: No

Reviewer #2: Yes

Reviewer #3: Yes

5. Review Comments to the Author

Reviewer #1: I would like to congratulates the authors for having successfully conducted a well-prepared RCT. This is a cross topic involving anesthesia and vascular surgery, and it seems that few studies to date have investigated this topic. Though negative results yielded, the study is still considered of good value for understanding the impact of magnesium on vascular surgery.

I have the following questions and suggestions:

1. The identifier of trial registration is different in title page(jRCTs041190013) and Methods and Materials (iRCTs041190013), please check.

Response: Thank you for the important point for our manuscript. “jRCT” is correct. We rewrote this. 

2. At page 5, “The first patient entered the study from 26 October 2018 to 15 July 2020. ” caused confusion, please check and revise.

Response: Thank you for the important point for our manuscript. We deleted the 15 July 2020 for avoiding confusion.

3. At page 5, “A computer-generated randomization programme was used. ”. This should be described in detail, what software did you use? Or what is the block size of randomization?

Response: Thank you for the important point for our manuscript. Actually, we used the research randomizer from on the Web, which was used 35.2 billion sets of random numbers for research study since 2007. We added this information in the text. 

4. From page 6 to 9, you described of performing anesthesia, performing surgery, and transferring patients to ICU. This part is too long, please make it concise but understandable.

Response: Thank you for the important point for our manuscript. We tried to make these sentences shorter.

5. At page 11, the determination of sample size is questionable. The data of sinus surgery and your unpublished data was combined to calculate sample size, which undoubtably caused imprecision. If this is not amendable, please explain in the limitation section.

Response: Thank you for the important point for our manuscript. We added the lower paragraph in the limitation section.

 In addition, our small simple size was calculated with our previous data that had showed many cases of delirium and in the interim analysis our study could have shown the statistical significance, however, finally, we could not show the statistical significance. We should have more careful calculation in sample size and should have taken more numbers in the sample size.

6. In the discussion section, you mentioned that the “magnesium target total of 60 mg•kg−1, actually 48±8 mg•kg−1, was not enough to obtain good outcomes ” and “Our protocol of magnesium administration was comparatively low. ”.

This could really be misleading, because your dosage is not low! Your dosage of “ 48±8 mg•kg−1” is actually higher that most magnesium trials [as in PMID: 17513654]. This rationale should be reconsidered.

Response: Thank you for the important point for our manuscript. And we changed those sentences are “Our protocol of magnesium administration may be comparatively low for obtaining good results. As you pointed out, our administration was not low, but we thought that to get good outcomes, it could make good outcomes that we should administer more dose of magnesium.

7. In the discussion section, you mentioned that the incidence of agitation and delirium were low. This contradicts to your sample size calculation. If you have known the agitation and delirium is rare, would you used your current sample size calculating method?

I understand that what’s done is hard to change, please think of a way to make this more logical.

Response: Thank you for the important point for our manuscript. Before the starting this study, we could not expect that incidence of delirium reduce. And in interim analysis of this study we could expect the statistical significance. However, we could not show the statistical significance. Therefore, we should point out the sample size may be small to detect the statistical significance. If we could know these things happen, we could have consider and recalculate the sample size and take larger sample size but we could not predict these things. We added in discussion section that “and the sample size might be small to detect statistical significance.”

8. The language is poor and not easy to understand, please seek professional assistance.

Response: Thank you for the important point for our manuscript. We asked the EDANZ to edit our manuscript and we wrote this in acknowledgements section.

Reviewer #2: This is an RCT looking at the effect of magnesium infusion on post-operative pain relief and sedative level after surgery for endovascular aortic aneurysm repair (EVAR).

1) It would be beneficial for the manuscript if the authors can give more details in the introduction. For example just mentioning that “effect on postoperative sedation and delirium is unclear” might not be sufficient. Consider what study designs have been used previously to justify using an RCT for the current study, this would be good to know. For example, there is mention of a retrospective study in the sample size calculation (although this is unpublished).

Response: Thank you for the important point for our manuscript. We are the most important clinical question is whether magnesium which have showed pain relief effect also has an effect on delirium. In addition, Recently it was reported to decrease postoperative agitation. Therefore we added the next sentences before “effect on postoperative sedation and delirium is unclear”.

Recently it was reported to decrease postoperative agitation,9 which might suggest that magnesium could have an effect on not only pain relief but also sedative level. To our knowledge, for the first time, that paper showed possibility of magnesium which has showed pain relief also has an effect on delirium and sedative level.

2) Additionally considering the target patient population is aged 60 and above, a reason/rationale in the introduction relating to the commonality of pain and delirium in this target group would add more context to the study.

Response: Thank you for the important point for our manuscript. We planned an intention to treat analysis for this study. We added the next sentences in the introduction section.

We chose the vascular surgery setting and older patient population not to decrease incidents of delirium.

In addition, we added the next sentences in the material and methods section.

The reason why we chose aged 60 years or older was not to decrease the incidence of delirium, 

3) The introduction should explicitly state the primary and secondary objectives.

Response: Thank you for the important point for our manuscript. We planned an intention to treat analysis for this study. We added the next sentences in the introduction section.

and the primary outcome was an effect on the RASS scores and the second outcomes were effects on the incidence of delirium, pain score, and length of ICU stay. 

4) Where safety outcomes considered?

Response: Thank you for the important point for our manuscript. We stated that “During the observational period, no adverse events caused by magnesium infusion were recorded.” as last sentence in the result section.(page 13, line 13). We added the next clauses to this sentence.

“ , which were, for example, arrythmia, bradycardia, hypotension, effects of muscle paralysis, convulsion and so on.

5) Can authors also mention how missing data will be handled.

Response: Thank you for the important point for our manuscript. We planned an intention to treat analysis for this study. We added the next sentences in introduction section. Actually, there were no missing data in our study.

We planned to perform a double-blind, randomized, controlled trial and analysis of intention to treat analysis.

6) It would be beneficial for the manuscript if the authors can define population of analysis (e.g ITT, per-protocol, safety)

Response: Thank you for the important point for our manuscript. We planned an intention to treat analysis for this study. We added the next sentences in introduction section. Actually, there were no missing data in our study.

We planned to perform a double-blind, randomized, controlled trial and analysis of intention to treat analysis.

7) Did that interim that was scheduled to take place happen and if it did, was this accounted for in the power calculation to account for the type I error, if the interim did/didnt happen can this be stated in the manuscript.

Response: Thank you for the important point for our manuscript. And in the discussion section we stated that “In the middle of the study, we conducted a planned interim analysis after gathering the data from 30 patients. At this time, we obtained RASS scores of -0.27 point less in the magnesium group (statistically significant, P=0.0321) and decided to continue the study until the planned number, 64 of the patients. After completion of the study, we recalculated the number of cases necessary for obtaining statistical significance and the power of this setting. Using our results, the difference of mean RASS score was only 0.26, with the number needed to reach statistical significance being 109 cases for each group.”(page 17, line 7-14).

 In addition, we stated that “We gave up adding cases because the difference in RASS score of less than 0.5 would have no clinical meaning and each group containing 109 would be too difficult to accomplish. The power of this study was just 0.416. We may have to plan another setting in which to detect a statistically positive neurological effect by magnesium.” (page 17, line 14-18)

We hoped these sentences answered your questions. 

8) For randomisation, explicitly state that it was 1:1 ratio.

Response: Thank you for the important point for our manuscript. And we added it in the materials and methods section. (page 6, line 8)

and randomization was 1:1 ratio.

9) Although secondary outcomes have stated, adding information relating to repeated measures would add more information.

Response: Thank you for the important point for our manuscript. As for complications, we stated that “During the observational period, no adverse events caused by magnesium infusion were recorded.” as last sentence in the result section.(page 13, line 13). We added the next clauses to this sentence.

“ , which were, for example, arrythmia,bradycardia, hypotension, effects of muscle paralysis, convulsion and so on. (page 13, line 14)

And as for the defined period of the secondary outcomes, we added the taking scores with a period.

Secondary outcomes were incidence of delirium defined using CAM-ICU, pain according to NRS, use of analgesic drugs, which were during ICU stay and length of ICU stay. (page 10, line 11) 

10) The choice of words may need to be considered, i.e secondary outcomes as incidence, implies you are looking at incidence in the context of epidemiology (defined as number of cased in a defined period). This would mean you would need to report incidence per person time follow-up. Which this is not the case. Perhaps consider revising this.

Response: Thank you for the important point for our manuscript. And as for the defined period of the secondary outcomes, we added the taking scores with a period.

Secondary outcomes were incidence of delirium defined using CAM-ICU, pain according to NRS, use of analgesic drugs, which were during ICU stay and length of ICU stay. (page 10, line 11) 

11) Can the final couple of sentences for sample size explicitly state which SD was considered.

Response: Thank you for the important point for our manuscript. We stated the SD we used were in the statistical analysis section. “In this previous study, the SD was not presented. In another retrospective observational study (not published), we investigated the rate of hyperactive delirium among patients who underwent EVAR surgery under general anesthesia in our hospital in the same setting. We reported RASS scores of 0.25±0.5 and 58 cases of hyperactive delirium among 237 patients. Therefore, we estimated that score levels differed by 1 to 2 with SD of 0.5 to 1. Assuming an α level of 0.05 and 90% power, the required number of patients for each group to observe a RASS difference is 22 at most. We considered that a total workable sample size would be 30 patients to detect a difference of 1 to 2, assuming a two-tailed type I error of 5% and type II error of 10%. To allow for a 5% dropout rate, we randomly assigned 32 patients.

 We hoped these sentences answered your questions. 

12) The analysis is based on multiple time-points how was this accounted for in the analysis, multiple testing.

Response: Thank you for the important point for our manuscript. We took the analysis based on multiple time-points, most of other similar papers take the same statistically method like our study. Therefore we took this method,

Reviewer #3: Well conducted randomized study looking at the effects of magnesium on RASS, NPS, and CAM-ICU.

Although negative studies are valuable to the literature, this negative study is difficult to really incorporate into our understanding of the effects of magnesium. The study size was very small and as the authors point out, the power of this study was just 0.416. This also is only a single center study looking at a single surgical case with relatively low pain issues. It doesn't seem that the magnesium contributed to any changes in intra-operative or post-operative narcotic use.

Response: Thank you for the important point for our manuscript. 

Maybe, as you pointed out, it doesn't seem that the magnesium contributed to any changes in intra-operative or post-operative narcotic use. However, there are a few positive effects on narcotic use. We hope more lager mount has a positive effect. 

6. PLOS authors have the option to publish the peer review history of their article (what does this mean?). If published, this will include your full peer review and any attached files.

---

## [Decision Letter · Decision Letter 1]

1 Dec 2022

PONE-D-22-03920R1Effect of intraoperative systemic magnesium sulphate on postoperative Richmond Agitation-Sedation Scale score after endovascular repair of aortic aneurysm: a double-blind, randomized, controlled trialPLOS ONE

Dear Dr. Fujita,

Thank you for submitting your manuscript to PLOS ONE. After careful consideration, we feel that it has merit but does not fully meet PLOS ONE’s publication criteria as it currently stands. Therefore, we invite you to submit a revised version of the manuscript that addresses the points raised during the review process. I believe that you have adressed the reviewer's comments but I would strongly like to see the following improvements:-please make sure you do not detail too much your methods in the introduction-please describe the timing of evaluation of the RASS. My feeling is that "at ICU transfer" is not detailed enough. As this is the primary outcome I believe it is important to be more specific. Did you evaluate the RASS 2 hours after the end of surgery? Is the patient immediately transfered in the ICU or after extubation in the PACU?-please be more specific in the results about the edverse events: I believe that "and so on" is too blurry.-please shorten your discussion. I believe that some paragraph are too putative to be discussed (level of blood Mg to be reached) or the paragraph before the limitations has too little evidence in the literature to be discussed.

Overall, I believe that your manuscript has reached high quality and my concerns can be easily addressed.

We look forward to receiving your revised manuscript.

Kind regards,

Raphael Cinotti, MD, PhD

Academic Editor

PLOS ONE

Journal Requirements:

Reviewers' comments:

Reviewer's Responses to Questions

**Comments to the Author**

1. If the authors have adequately addressed your comments raised in a previous round of review and you feel that this manuscript is now acceptable for publication, you may indicate that here to bypass the “Comments to the Author” section, enter your conflict of interest statement in the “Confidential to Editor” section, and submit your "Accept" recommendation.

Reviewer #2: All comments have been addressed

Reviewer #3: (No Response)

Reviewer #4: (No Response)

2. Is the manuscript technically sound, and do the data support the conclusions?

Reviewer #2: Yes

Reviewer #3: No

Reviewer #4: Yes

3. Has the statistical analysis been performed appropriately and rigorously? 

Reviewer #2: Yes

Reviewer #3: No

Reviewer #4: Yes

4. Have the authors made all data underlying the findings in their manuscript fully available?

Reviewer #2: No

Reviewer #3: Yes

Reviewer #4: Yes

5. Is the manuscript presented in an intelligible fashion and written in standard English?

Reviewer #2: Yes

Reviewer #3: No

Reviewer #4: Yes

6. Review Comments to the Author

Reviewer #2: (No Response)

Reviewer #3: The study size was very small. There was not enough power to determine the effect of magnesium and postoperative delirium and agitation. It really is difficult to make any useful conclusion from this study.

Unfortunately, at the same time, the goal magnesium of 60 mg/kg was not reached. From the data, this goal looks like it was never likely to be reached based on anesthesia time and the study protocol of 30 mg/kg for the first hour and then 10 mg/kg for each hour after. The maximum anesthesia time was only ~200 minutes and some cases were as short as 170 minutes.

The anesthetic protocol seems quite varied. Induction of doses of propofol and remifentanil were several fold different. There was no mention of their anesthetic targets. And postoperative use of fentanyl and dexmedetomidine were not described in the methods. While the amount of fentanyl and incidence of dexmedetomidine use look similar between groups, it is unclear if this could have affected the number of cases of delirium and agitation.

The authors state BP goals were to maintain BP from 70% and 130%, but then nicardipine was used to treat hypertension, which was more than 140 mmHg of systolic blood pressure. It would seem for many patients 140 mmHg is not close to 130% of normal BP. This is confusing.

There is something wrong with this sentence: U10, linese of acetaminophen and dexmedetomidine,

Page 13, paragraph 2: looks like magnesium levels are reported first for the magnesium group and then the control group except for the ionized magnesium at postoperative ICU transfer. This is confusing.

Not sure this conclusion is determined by the study: To obtain a neurological effect, we should follow a higher-dose protocol than that employed in the present study. It is a speculation made by the authors.

Table 2: maybe specify that Postoperative ICU transfer is time 0 hr

Manuscript needs editing for clarity.

Reviewer #4: I would like to complement the authors with this study that, although small and negative, is important and well done. The primary outcome was whether magnesium had an effect on RASS score of patients at postoperative ICU transfer. No effect observed.

Minor comments:

Change titel to:

Effect of intraoperative systemic magnesium sulphate on postoperative Richmond Agitation-Sedation Scale score after endovascular repair of aortic aneurysm UNDER GENERAL ANAESTHESIA: a doubleblind, randomized, controlled trial Short

Change short title to:

Effect of i.v. magnesium on RASS score after EVAR UNDER GENERAL ANAESTHESIA

The primary outcome was whether magnesium had an effect on RASS score of patients at postoperative ICU transfer. No effect observed. Likely because the study is small and therefore underpowered to detect meaningful differences

Methods: the authors state “ All P values are two-tailed.” Which is not exactly the way to formulate this. All statistical test were performed two sided.

Results:

“ During the observational period, no adverse events caused by magnesium infusion were

recorded, which were, for example, arrythmia, bradycardia, hypotension, effects of

muscle paralysis, convulsion and so on.” Remove the “and so on”

Discussion:

“ First, an administered magnesium target total of 60 mg•kg−1, actually 48±8 mg•kg−1, was

not enough to obtain good outcomes” All otcomes are “ good outcomes” I think you mean that the Mg dose used might be to low tom reduce the postoperative delirium in patients undergoing EVAR surgery under general anesthesia

Suppl figures:

Remove the operation data fro your excel file… otherwise the participants arte not fully anaonimsed ( age surgery date and hospital could lead to a traceble study subject)

Suppl figure: Click here to access/download Supporting Information renamed_3b3df.docx Is only in jappanese state that in the suppl figure text. The Japanese supple materials can not be checked by me.

7. PLOS authors have the option to publish the peer review history of their article (what does this mean?). If published, this will include your full peer review and any attached files.

Reviewer #2: No

Reviewer #3: No

Reviewer #4: **Yes: **Matijs van Meurs MD PhD

---

## [Author Response · Author response to Decision Letter 1]

8 Jan 2023

Response to comments of the academic editor:

1, -please make sure you do not detail too much your methods in the introduction

Response: Thank you for the important point for our manuscript. We tried to rewrite the content according to your suggestions.

We deleted “We chose the vascular surgery setting and older patient population not to decrease incidents of delirium. We planned to perform a double-blind, randomized, controlled trial and analysis of intention to treat analysis. We investigated its effect on the postoperative Richmond Agitation-Sedation Scale (RASS) score, pain score (numerical rating scale: NRS), delirium (Confusion Assessment Method for the Intensive Care Unit: CAM-ICU), and length of ICU stay.” And the word explanations of abbreviations were added to the following the sentences.

2. -please describe the timing of evaluation of the RASS. My feeling is that "at ICU transfer" is not detailed enough. As this is the primary outcome I believe it is important to be more specific. Did you evaluate the RASS 2 hours after the end of surgery? Is the patient immediately transferred in the ICU or after extubation in the PACU?

Response: Thank you for the important point for our manuscript. These were not detailed enough. We tried to rewrite the content according to your suggestions.

Actually, all patients were extubated at the operating room and were moved to the ICU. And, as soon as the patients arrived at the ICU, the ICU nurses examined vital signs and evaluated RASS, NRS and CAM-ICU. And then, we evaluated the patients 1 hr, 6 hr and 24 hr after the admission of ICU. 

Therefore, we added the sentence “all patients were extubated at the operating room and were moved to the ICU. And, as soon as the patients arrived at the ICU, the ICU nurses examined vital signs and evaluated RASS, NRS and CAM-ICU. 

In addition, we use the words “ICU admission” instead of “ICU transfer”.

3. -please be more specific in the results about the adverse events: I believe that "and so on" is too blurry.

Response: Thank you for the important point for our manuscript. These sentences were not detailed enough. We tried to rewrite the content according to your suggestions.

Actually, we examined existence or non-existence of arrythmia, bradycardia, hypotension, effects of muscle paralysis, and convulsion. Therefore, we rewrote 

“During the observational period, no adverse events caused by magnesium infusion were recorded, which were existence or non-existence of arrythmia, bradycardia, hypotension, effects of muscle paralysis, and convulsion.”

4. -please shorten your discussion. I believe that some paragraph are too putative to be discussed (level of blood Mg to be reached) or the paragraph before the limitations has too little evidence in the literature to be discussed.

Response: Thank you for the important point for our manuscript. We tried to shorten discussion and rewrite the content according to your suggestions.

We deleted many sentences and two paragraph in discussion section to be easy to understand our paper and we could reduce the word counts to 3,170 from 3,620. 

Response to comments of ‘Journal Requirements:

Response: Thank you for the important point for our manuscript. We reviewed our references. And we confirmed if our cited papers are retracted. We could not find any retracted papers in our references with PubMed.gov on the WEB.

Response to comments of the reviewers:

1. If the authors have adequately addressed your comments raised in a previous round of review and you feel that this manuscript is now acceptable for publication, you may indicate that here to bypass the “Comments to the Author” section, enter your conflict of interest statement in the “Confidential to Editor” section, and submit your "Accept" recommendation.

Reviewer #2: All comments have been addressed

Reviewer #3: (No Response)

Reviewer #4: (No Response)

2. Is the manuscript technically sound, and do the data support the conclusions?

Reviewer #2: Yes

Reviewer #3: No

Reviewer #4: Yes

3. Has the statistical analysis been performed appropriately and rigorously?

Reviewer #2: Yes

Reviewer #3: No

Reviewer #4: Yes

4. Have the authors made all data underlying the findings in their manuscript fully available?

Reviewer #2: No

Reviewer #3: Yes

Reviewer #4: Yes

5. Is the manuscript presented in an intelligible fashion and written in standard English?

Reviewer #2: Yes

Reviewer #3: No

Reviewer #4: Yes

6. Review Comments to the Author

Reviewer #2: (No Response)

Reviewer #3: The study size was very small. There was not enough power to determine the effect of magnesium and postoperative delirium and agitation. It really is difficult to make any useful conclusion from this study.

Response: Thank you for the important point for our manuscript. And we could not show that Mg was effective for postoperative delirium and agitation. However, we planned a prospective RCT with calculating proper sample size. And we reached the conclusion that we could not obtain good results with the method of our protocol in this paper and we think that that is informative results in the field of Mg research.

Unfortunately, at the same time, the goal magnesium of 60 mg/kg was not reached. From the data, this goal looks like it was never likely to be reached based on anesthesia time and the study protocol of 30 mg/kg for the first hour and then 10 mg/kg for each hour after. The maximum anesthesia time was only ~200 minutes and some cases were as short as 170 minutes.

Response: Thank you for the important point for our manuscript. And we could not control the length of the operation. Therefore, we could not show avoid the cases with small amount of Mg. However, the same things might happen in clinical settings and measured ionized Mg had statistical differences between control group and Mg group. Next time, we have to consider that the pointed-out things would happen.

The anesthetic protocol seems quite varied. Induction of doses of propofol and remifentanil were several fold different. There was no mention of their anesthetic targets. And postoperative use of fentanyl and dexmedetomidine were not described in the methods. While the amount of fentanyl and incidence of dexmedetomidine use look similar between groups, it is unclear if this could have affected the number of cases of delirium and agitation.

Response: Thank you for the important point for our manuscript. In clinical setting, we thought that our results with amounts of anesthetic drugs were usual in normal anesthetic managements. And postoperative uses of fentanyl and dexmedetomidine were shown in table 4. We concluded that with these data, we reached our conclusion. 

The authors state BP goals were to maintain BP from 70% and 130%, but then nicardipine was used to treat hypertension, which was more than 140 mmHg of systolic blood pressure. It would seem for many patients 140 mmHg is not close to 130% of normal BP. This is confusing.

Response: Thank you for the important point for our manuscript. We used nicardipine for 140mmHg of systolic blood pressure if 140 mmHg was within 130% of its normal BP for the patient.

There is something wrong with this sentence: U10, linese of acetaminophen and dexmedetomidine,

Response: We are really sorry for misspelling. We have rewritten this sentence. 

Page 13, paragraph 2: looks like magnesium levels are reported first for the magnesium group and then the control group except for the ionized magnesium at postoperative ICU transfer. This is confusing.

Response: We are really sorry for misspelling. We have rewritten this sentence. We always reported the magnesium group first.

Not sure this conclusion is determined by the study: To obtain a neurological effect, we should follow a higher-dose protocol than that employed in the present study. It is a speculation made by the authors.

Response: Thank you for the important point for our manuscript. It is our speculation. Therefore, we should weaken the tone of these sentences about using a higher dose. In discussion section, we have rewritten this sentence. 

We changed “To obtain a neurological effect, we should follow a higher-dose protocol than that employed in the present study” into “We might have to follow a higher-dose level of ionized magnesium to demonstrate a favorable neurological effect.”

Table 2: maybe specify that Postoperative ICU transfer is time 0 hr

Manuscript needs editing for clarity.

Response: Thank you for the important point for our manuscript. We have rewritten this sentence. 

We changed “There was no significant effect on RASS score (0 [0 to 0] vs 0 [0 to 0]; P=0.114) at postoperative ICU tranfer (Table 2). “ into “There was no significant effect on RASS score (0 [0 to 0] vs 0 [0 to 0]; P=0.114) at postoperative ICU admission, time 0 hour (Table 2). In addition, in table 2, we added “(0h)”after ICU admission.

Reviewer #4: I would like to complement the authors with this study that, although small and negative, is important and well done. The primary outcome was whether magnesium had an effect on RASS score of patients at postoperative ICU transfer. No effect observed.

Minor comments:

Change title to:

Effect of intraoperative systemic magnesium sulphate on postoperative Richmond Agitation-Sedation Scale score after endovascular repair of aortic aneurysm UNDER GENERAL ANAESTHESIA: a doubleblind, randomized, controlled trial Short

Change short title to:

Effect of i.v. magnesium on RASS score after EVAR UNDER GENERAL ANAESTHESIA

Response: Thank you for the important point for our manuscript. We added “under general anaesthesia” to the titles. 

The primary outcome was whether magnesium had an effect on RASS score of patients at postoperative ICU transfer. No effect observed. Likely because the study is small and therefore underpowered to detect meaningful differences

Response: Thank you for the important point for our manuscript. And we could not show that Mg was effective for postoperative delirium and agitation. However, we planned a prospective RCT with calculating proper sample size. And we reached the conclusion that we could not obtain good results with the method of our protocol in this paper and we think that that is informative results in the field of Mg research.

Methods: the authors state “ All P values are two-tailed.” Which is not exactly the way to formulate this. All statistical test were performed two sided.

Response: Thank you for the important point for our manuscript. We changed “All P values are two-tailed” to “All statistical test were performed two sided” . 

Results:

“ During the observational period, no adverse events caused by magnesium infusion were

recorded, which were, for example, arrythmia, bradycardia, hypotension, effects of

muscle paralysis, convulsion and so on.” Remove the “and so on”

Response: Thank you for the important point for our manuscript. These sentences were not detailed enough. We removed “and so on”.

Discussion:

“ First, an administered magnesium target total of 60 mg•kg−1, actually 48±8 mg•kg−1, was not enough to obtain good outcomes” All outcomes are “ good outcomes” I think you mean that the Mg dose used might be to low tom reduce the postoperative delirium in patients undergoing EVAR surgery under general anesthesia

Response: Thank you for the important point for our manuscript. It is our speculation. Therefore, we should weaken the tone of these sentences about using a higher dose. In discussion section, we have rewritten this sentence. 

We changed “First, an administered magnesium target total of 60 mg•kg−1, actually 48±8 mg•kg−1, was not enough to obtain good outcomes” into “First, an administered magnesium target total of 60 mg•kg−1, actually 48±8 mg•kg−1, might not be enough to obtain good outcomes”

Suppl figures:

Remove the operation data from your excel file… otherwise the participants are not fully anaonimsed (age surgery date and hospital could lead to a traceble study subject)

Response: Thank you for the important point for our manuscript. Another reviewer referred to the operation management. Therefore, we remained the operation data. 

Suppl figure: Click here to access/download Supporting Information renamed_3b3df.docx Is only in jappanese state that in the suppl figure text. The Japanese supple materials can not be checked by me.

Response: We are really sorry for your inconvenience. We showed the original version in supporting information in original language, Japanese. You can read the protocol in English version in jRCT protocol on the WEB. In addition, you can see the file “the translation of the main points of our protocol” in English in supporting information. 

7. PLOS authors have the option to publish the peer review history of their article (what does this mean?). If published, this will include your full peer review and any attached files.

Do you want your identity to be public for this peer review? For information about this choice, including consent withdrawal, please see our Privacy Policy.

Reviewer #2: No

Reviewer #3: No

Reviewer #4: Yes: Matijs van Meurs MD PhD

Response: Thank you, professor Matijs van Meurs. We really appreciated that you took your time to review our paper.

---

## [Editor Report · Decision Letter 2]

24 Jan 2023

Effect of intraoperative systemic magnesium sulphate on postoperative Richmond Agitation-Sedation Scale score after endovascular repair of aortic aneurysm under general anesthesia: a double-blind, randomized, controlled trial

PONE-D-22-03920R2

Dear Dr. Fujita,

We’re pleased to inform you that your manuscript has been judged scientifically suitable for publication and will be formally accepted for publication once it meets all outstanding technical requirements.

Kind regards,

Raphael Cinotti, MD, PhD

Academic Editor

PLOS ONE
---

## [Editor Report · Acceptance letter]

27 Jan 2023

PONE-D-22-03920R2 

Effect of intraoperative systemic magnesium sulphate on postoperative Richmond Agitation-Sedation Scale score after endovascular repair of aortic aneurysm under general anesthesia: a double-blind, randomized, controlled trial 

Dear Dr. Fujita:

I'm pleased to inform you that your manuscript has been deemed suitable for publication in PLOS ONE. Congratulations! Your manuscript is now with our production department. 

Kind regards, 

on behalf of

Pr. Raphael Cinotti 

Academic Editor

PLOS ONE